# Rolling Circle cDNA Synthesis Uncovers Circular RNA Splice Variants

**DOI:** 10.3390/ijms20163988

**Published:** 2019-08-16

**Authors:** Aniruddha Das, Pranita K. Rout, Myriam Gorospe, Amaresh C. Panda

**Affiliations:** 1Institute of Life Sciences, Nalco Square, Bhubaneswar 751023, India; 2School of Biotechnology, KIIT University, Bhubaneswar 751023, India; 3Laboratory of Genetics and Genomics, National Institute on Aging, NIH, Baltimore, MD 21224, USA

**Keywords:** circRNAs, backsplice sequence, divergent primers, RNase R, RT-PCR, alternative splicing, splice variants

## Abstract

High-throughput RNA sequencing and novel bioinformatic pipelines have identified thousands of circular (circ)RNAs containing backsplice junction sequences. However, circRNAs generated from multiple exons may contain different combinations of exons and/or introns arising from alternative splicing, while the backsplice junction sequence is the same. To be able to identify circRNA splice variants, we developed a method termed circRNA-Rolling Circle Amplification (circRNA-RCA). This method detects full-length circRNA sequences by performing reverse transcription (RT) in the absence of RNase H activity, followed by polymerase chain reaction (PCR) amplification of full-length circRNAs using a forward primer spanning the backsplice junction sequence and a reverse primer exactly upstream of the forward primer. By sequencing the PCR products, circRNA splice variants bearing the same backsplice junctions, which were otherwise only predicted computationally, could be experimentally validated. The splice variants were further predicted to associate with different subsets of target RNA-binding proteins and microRNAs, supporting the notion that different circRNA splice variants can have different biological impacts. In sum, the circRNA-RCA method allows the accurate identification of full-length circRNA sequences, offering unique insight into their individual function.

## 1. Introduction

Circular (circ)RNAs are a large family of covalently closed RNA molecules expressed ubiquitously in various organisms, including humans [1,2]. Recent high-throughput RNA sequencing (RNA-seq) analyses have identified thousands of circRNAs, which can range in size from <100 nucleotides to several thousand nucleotides [3]. The majority of reported circRNAs are derived from exons of precursors (pre) of messenger RNAs (mRNAs) and noncoding (nc)RNAs by a process called backsplicing [3,4,5,6,7,8]. Additionally, circRNAs can also be generated from introns and from both introns and exons [9,10,11]. Due to a lack of free ends, circRNAs are resistant to exonuclease activity and are believed to be highly stable in cells [8,12]. Although more than one hundred thousand circRNAs have been identified to date, the functions of circRNAs have only been elucidated for a handful of them; these functions often include sequestration of microRNAs (miRNAs) and RNA-binding proteins (RBPs), as well as competition with linear RNA splicing [7,8,13,14].

The unique backsplice junction sequence in the circRNA serves as a key feature for their identification and quantification [2,15]. The mature spliced sequences of circRNAs are bioinformatically predicted considering the sequences encompassed between the backsplice junction coordinates [3,11]. However, multiple circRNA splice variants with the same backsplice sites can be generated from the same gene locus by alternative selection of exons and introns during circRNA biogenesis [16,17,18]. Recent developments in circRNA enrichment methods and high-throughput RNA-seq methodologies have led to the genome-wide identification of circRNA splice variants in a few model organisms [11,16,17,18]. Since the body of a mature circRNA is also found in the counterpart linear RNA and the levels of the circRNA are often lower than the levels of the linear RNA counterpart, it is technically challenging, expensive, and often error-prone to derive full-length circRNA sequences directly from RNA-sequencing data.

Furthermore, given that the function of a circRNA is believed to depend on the mature sequence of the circRNA, sequence differences in the mature sequence could result in differential functions by circRNA splice variants [14]. Here, we describe a novel method to identify circRNA splice variants with the same backsplice sequence. Total RNA was first enriched for circRNA by digesting with RNase R, whereupon tandem repeats of complementary (c)DNA were synthesized with reverse transcriptase without RNase H activity. The specific circRNA amplicons were generated using special primers, with one primer spanning the backspliced junction and the reverse primer exactly upstream of the forward primer, allowing the full-length amplification of circRNA. As proof-of-principle, we have identified circRNA splice variants with the same backsplice sequence in human cervical carcinoma HeLa cells and have predicted their association with microRNAs and RBPs.

## 2. Results

### 2.1. Identification of circRNA Backsplice Sequences in HeLa Cells

RNA-seq analysis of circRNAs from HeLa cells identified many circRNAs with mature sequences that differed from those previously reported [11]. We selected a few abundant circRNAs identified to have splice variants in HeLa cells to validate the expression. We amplified a few select circRNAs using specific divergent primers that were designed to amplify the circRNA backsplice junction sequence (Figure 1A). Following the amplification of the corresponding junction sequences from HeLa whole-cell RNA, the PCR products were resolved by electrophoresis on 2% agarose gels stained with SYBR Gold (Figure 1B) and were subsequently purified and sequenced to confirm the backsplice sequences (Figure 1C; Appendix A) [19]. To identify true circRNAs in HeLa cells, select circRNAs were analyzed for RNase R resistance [12,19]. As expected, the circRNAs tested were resistant to digestion with RNase R, while the linear RNAs were degraded (Figure 1D). 

### 2.2. Identification of circRNA Splice Variants with Identical Backsplice Sequence by RT-PCR

As previously reported [11], many circRNAs have isoforms with the same backsplice sequence and altered exon inclusion/deletion in the mature circRNA sequence. Thus, we sought to identify the full-length sequence of select circRNAs in HeLa cells by generating tandem-repeat cDNA amplicons of the full-length circRNA, by performing rolling circle reverse transcription (RT) using RNase H-minus reverse transcriptase followed by PCR amplification using full-length primers (Figure 2). The Sanger sequencing of the PCR products amplified from the above reaction revealed circRNA splice variants in HeLa cells (Figure 2). 

For instance, we identified two splice variants of *hsa_circ_0007127* measuring 266 nt and 324 nt in length, while the 490-nt isoform reported by circBase was not detected [3] (Figure 3; Figure 4A,B). In another example, the mature sequence of *hsa_circ_0001566* was predicted by circBase to span 497 nt [3], while a splice variant was detected to be 738 nt long (Figure 3; Appendix A). We also identified splice variants of *hsa_circ_0009156, hsa_circ_0071410, hsa_circ_0084615*, and *hsa_circ_0007822* (Figure 3; Table 1). Interestingly, we could only verify the expression of the circBase reported 264-nt isoform of *hsa_circ_0003964* and could not successfully validate the sequence of the 535-nt splice variant that was predicted from the RNA-seq data previously [11]. However, the sequencing of the larger products amplified by PCR analysis for *hsa_circ_0009156, hsa_circ_0003964, hsa_circ_0007822, hsa_circ_0007127,* and *hsa_circ_0084615* did not reveal meaningful sequences due to either their sequence heterogeneity or their overlap with other bands. However, these PCR products could be doublets (two rounds of full-length circRNA amplification) of identified circRNAs or another circRNA isoform or a nonspecific product. Since we purified and sequenced specific clean bands amplified by the circRNA-RCA procedure, other nonspecific bands in the gel did not interfere with the overall analysis. Together, our circRNA-RCA method identified the expression of circRNA splice variants with or without the expression of the circRNA isoform reported in circBase.

### 2.3. Identification of Actual Full-Length Sequence and Exon Composition of circRNA Splice Variants 

The mature spliced sequence of a circRNA may vary due to alternative splicing of exons/introns during circRNA biogenesis [16,17]. According to circBase, the spliced *hsa_circ_0007127* was reported to be 490 nt generated from exons 2, 3, and 4 of the *CNOT2* precursor-mRNA (pre-mRNA) transcript (NR_037615.1), which was not detected by circRNA-RCA analysis [3] (Table 1). Interestingly, however, sequencing the PCR products of *hsa_circ_0007127* revealed two splice variants, one 266 nt long, containing exons 2 and 4, and another one 324 nt long, containing exons 2, 4, and part of exon 3 (Figure 4A, B). Surprisingly, the circRNA-RCA method identified a novel 324-nt splice variant of *hsa_circ_0007127*, while the short 266-nt variant was previously predicted from RNA-seq data [11]. Sequencing the full-length PCR products of *hsa_circ_0001566* revealed the inclusion of a 241-nt novel exon between the annotated exon 2 and exon 3 of MAPK9 pre-mRNA (NM_139069) (Appendix A). The inclusion of this extra sequence led to the biogenesis of an *hsa_circ_0001566* splice variant that was 738 nt long (Appendix A). In addition, we identified the expression of splice variants for *hsa_circ_0007822* (Appendix A), *hsa_circ_0084615* (Appendix A), *hsa_circ_0009156* (Appendix A), and *hsa_circ_0071410* (Appendix A) in HeLa cells. These findings support the notion that circRNA splice variants arising from distinct combinations of introns and exons are highly prevalent.

### 2.4. Differential Association of miRNAs and RBPs with circRNA Splice Variants 

Given the presence of different sequences in the full-length transcripts of circRNA splice variants, the presence of binding sites and hence their putative association with target microRNAs and RBPs also differed among variants. To assess differences in the subsets of microRNAs associated with circRNA splice variants, we used the miRDB web tool [20]. As shown in Table 2, the short and long splice variants of *hsa_circ_0007127* were predicted to target seven and nine miRNAs respectively, while the sequence predicted by circBase included 13 miRNAs. 

Additionally, we predicted the differential association of miRNAs with different splice variants of *hsa_circ_0009156* (Appendix A). Our computational analysis did not predict any differential association of miRNAs with the splice variants of *hsa_circ_0084615, hsa_circ_0007822*, or *hsa_circ_0071410* (data not shown). Furthermore, analysis of RBPs binding to *hsa_circ_0009156* and *hsa_circ_0084615* using the RBPmap web tool revealed that the circRNA splice variants are differentially associated with several RBPs as compared with those predicted using the sequences provided by circBase [21] (Appendix A). Collectively, these findings indicate that the experimental identification of mature full-length circRNA sequences is critical for the prediction of the RBPs and microRNAs associating with the mature sequences of circRNAs.

### 2.5. Potential Effect of circRNA Splice Variants on Gene Expression

CircRNA splice variants are expected to regulate different biological processes, at least in part, through their altered association of microRNAs and RBPs. To predict the biological functions of *hsa_circ_0007127* splice variants, we first identified the set of experimentally validated target genes (miRTarBase) of miRNAs associated with the reported circBase sequence and the circRNA splice variants, using the miRWalk web tool [22]. Using this web tool, we analyzed the association of these genes with different biological processes (BP) in the Gene Ontology (GO) database for *hsa_circ_0007127* splice variants (Appendix A). Interestingly, computational analysis suggested different biological processes influenced by the microRNAs bound to the long and short splice variants of *hsa_circ_0007127* compared with the pathways affected by *hsa_circ_0007127* as defined by circBase (Figure 5). Gene set enrichment analysis revealed that several biological processes enriched for the experimental and predicted circRNAs were shared, including “viral process”, “cell cycle”, “histone mRNA metabolic process”, and “spliceosomal snRNP assembly”. However, some biological processes, such as “response to cytokine”, “negative regulation of translation”, and “response to hydrogen peroxide” (Figure 5), were exclusively enriched for the circBase-reported and long splice variant of *hsa_circ_0007127*. As expected, some biological processes, such as “response to iron ion”, “intracellular transport of virus”, and “apoptotic DNA fragmentation” were exclusively enriched for the miRNAs targeting the circBase-reported sequences of *hsa_circ_0007127.*

Furthermore, we predicted the differential association of different biological processes by miRNAs associated with splice variants of *hsa_circ_0009156* (data not shown). In sum, the altered association of miRNAs with different circRNA splice variants is expected to lead to differential regulation of target genes involved in various key cellular pathways.

## 3. Discussion

Three decades have elapsed since the initial discovery of circRNAs, but only the arrival of high-throughput RNA-sequencing and advanced computational tools has begun to enable studies of the sequences, abundance, and function of circRNAs in various organisms. Progress has been slow, however, given that the methods to identify and characterize circRNAs are still rudimentary and still rely on methods designed for linear RNAs. For this reason, the functions of only a handful of circRNAs have been studied among the tens of thousands of circRNAs that have been annotated. To begin to fill the void in circRNA-specific methodologies, we present a novel method designed to identify circRNA variants that share the same junction sequence but have different circRNA bodies.

Current methods to identify and quantify circRNAs, including circRNA sequencing, circRNA microarray, and northern blot and RT-qPCR analyses, all rely on the backsplice junction sequence of circRNA [8,11,13,23,24], as the backsplice junction sequence can be predicted with relatively high confidence, while the body of the circRNA can only be inferred. However, the full sequence of a mature circRNA after backsplicing may be altered in different cell types or tissues, as differential splicing events lead to the inclusion of distinct exons and/or introns and give rise to different circRNA splice variants [11,16,17,18]. The method described in this report aims to identify systematically and reliably the full-length sequence of circRNA variants that share the same backsplice junction sequence. The circRNA-RCA method begins with digestion with RNase R to enrich the sample in circRNA, followed by reverse transcription without RNase H activity to generate tandem repeats of cDNA templates for PCR amplification and/or sequencing (Figure 2). Forward PCR primers spanning 10 nt on each side of the backsplice junction of the circRNA are then used for specific amplification of circRNAs (Figure 2). The reverse primer is placed exactly upstream of the forward primer, allowing the amplification of full-length mature circRNA sequences. The amplification with the junction-spanning primer is highly specific to an individual circRNA, while the divergent primers can potentially amplify multiple circRNAs derived from the same precursor RNA. Importantly, the amplification of full-length circRNAs confirmed the existence of circRNA splice variants that share the same backsplice junction in HeLa cells. 

An altered selection of exon and/or intron sequences during circRNA biogenesis leads to changes in the production of mature spliced circRNAs. As circular RNAs are known to regulate gene expression by acting as decoys or sequestration factors for RBPs and miRNAs [14], we analyzed their interaction with miRNAs and RBPs using the miRDB and RBPmap web tools, respectively. As expected, circRNA splice variants were predicted to interact with different sets of miRNAs and RBPs (Table 2, Appendix A). Furthermore, the target mRNAs affected by the miRNAs binding to the circRNA splice variants are predicted to control different biological processes, underscoring the importance of identifying the actual mature sequences of circRNAs (Figure 5). 

Although our computational analysis suggested that different circRNA splice variants were potentially implicated in different cellular pathways, these predictions were based on miRDB and miRWalk only, and there are several other miRNA and pathway prediction algorithms available. The same limitation was true for RBP target site data. The expression of the predicted RBPs and miRNAs would then need to be experimentally validated in HeLa cells. Additionally, experimental validation is essential to verify the association of RBPs and miRNAs with circRNA splice variants and to elucidate their function. In summary, gaining deeper knowledge of circRNA splice variants using circRNA-RCA will allow for more accurate prediction of the molecules (microRNAs, RBPs, and likely other molecules) modulated by a given circRNA. In turn, this knowledge will enhance the potential usefulness of circRNAs as therapeutic targets, as well as diagnostic and prognostic biomarkers.

## 4. Materials and Methods 

### 4.1. Cell Culture and RNA Isolation

Human cervical carcinoma HeLa cells were cultured in Dulbecco’s modified Eagle’s medium (DMEM) containing 10% fetal bovine serum (FBS, Gibco) and 1% pen/strep (Thermo Fisher Scientific), in a humidified atmosphere of 95% air and 5% CO_2_, at 37 °C. Total RNA from the HeLa cells was isolated using the PureLink RNA Mini Kit (Thermo Fisher Scientific), following the manufacturer’s instructions.

### 4.2. Targets and PCR Primers

Several circRNAs with potential splice variants and higher expression levels in HeLa cells were selected from our previous publication (Table 1). The divergent (div) primers for circRNA detection and quantification were taken from previous publications or designed by the CircInteractome web tool [25]. The primers for full-length (fl) circRNA amplification were designed manually by placing the forward primer on the junction sequence spanning 10 nt on either side of the junction and placing the reverse primer exactly upstream to the forward primer. All oligomer sequences are provided in Appendix A. 

### 4.3. RNase R Treatment and cDNA Synthesis 

Five micrograms of total RNA was digested for 30 min at 37 °C with 1 μL of RNase R enzyme (Epicentre), followed by RNA isolation with the PureLink RNA isolation kit [19]. For cDNA synthesis, reverse transcription (RT) was performed using random hexamers and maxima reverse transcriptase (Thermo Fisher Scientific), following the manufacturer’s protocol [19]. For full-length circRNA cDNA, random primer was used with Maxima RNase H-minus reverse transcriptase (Thermo Fisher Scientific), following the manufacturer’s protocol. The cDNA reaction was incubated for 15 min at 37 °C with 1 μL of RNase H (New England Biolabs) before inactivating the RT enzyme. 

### 4.4. RT-PCR and circRNA Sequencing

RT followed by quantitative PCR (RT-qPCR) was performed using the Power Up SYBR Green master mix, following the manufacturer’s protocol (Thermo Fisher Scientific). PCR reaction settings included an initial step of 95 °C for 2 min, followed by 45 cycles of 95 °C for 2 s plus 60 °C for 5 s, followed by relative RNA level calculation, as described previously [11,19]. RT-PCR was performed with the full-length circRNA cDNAs and the full-length (fl) primers using the Dream Taq DNA polymerase with a cycle setup that included an initial step of 95 °C for 2 min followed by 40 cycles of 95 °C for 5 s, 58 °C for 20 s, and 72 °C for 60 s. The above PCR products were resolved on SYBR Gold stained agarose gel for visualization, followed by purification and Sanger sequencing to identify the mature circRNA sequences.

### 4.5. Target Predition and Functional Annotation of circRNAs

The sequences of reported circRNAs and circRNA splice variants were obtained from circBase (http://www.circbase.org/; accessed on 15 June 2018) and our previous publication [11], respectively. The potential miRNA targets of circRNAs were predicted using the custom prediction web tool in miRDB (http://mirdb.org; accessed on 5 December 2018) [20]. The mRNA targets of miRNAs predicted to target circRNAs were obtained from the miRWalk web tool, using the miRNA target search option with filters for 3’UTR target and miRTarBase (http://mirwalk.umm.uni-heidelberg.de/; accessed on 10 June 2019) [22]. The Gene Ontology (GO) annotations for biological processes for mRNA targets were obtained using the miRWalk web tool, using the gene set enrichment analysis option [22]. The association of RBPs with circRNAs were predicted using the RBPmap website (http://rbpmap.technion.ac.il/; accessed on 28 November 2018) [21]. 

## 5. Patents

An Indian Patent Application (no. 201931015071) filed on 15 April 2019 resulted from part of the work reported in this manuscript.

## Figures and Tables

**Figure 1 ijms-20-03988-f001:**
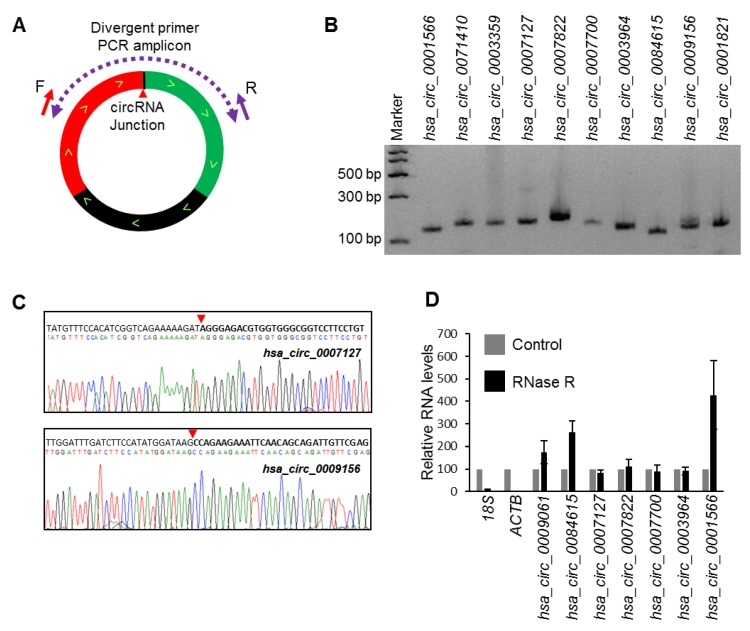
Validation of circRNA expression using divergent primers. (**A**) Schematic representation of the divergent primers used for detection and quantification of circRNAs. The red arrowhead represents the backsplice site. (**B**) PCR products amplified with divergent primers resolved on SYBR Gold-stained, 2% agarose gels. (**C**) Sanger sequencing of purified PCR products showing the backsplice junction sequences of mentioned circRNAs. The red arrowhead represents the backsplice site. (**D**) RT-qPCR (reverse transcription-qPCR) analysis of the resistance of circRNAs to RNase R digestion. Data represent the means ± S.E.M. from three experiments.

**Figure 2 ijms-20-03988-f002:**
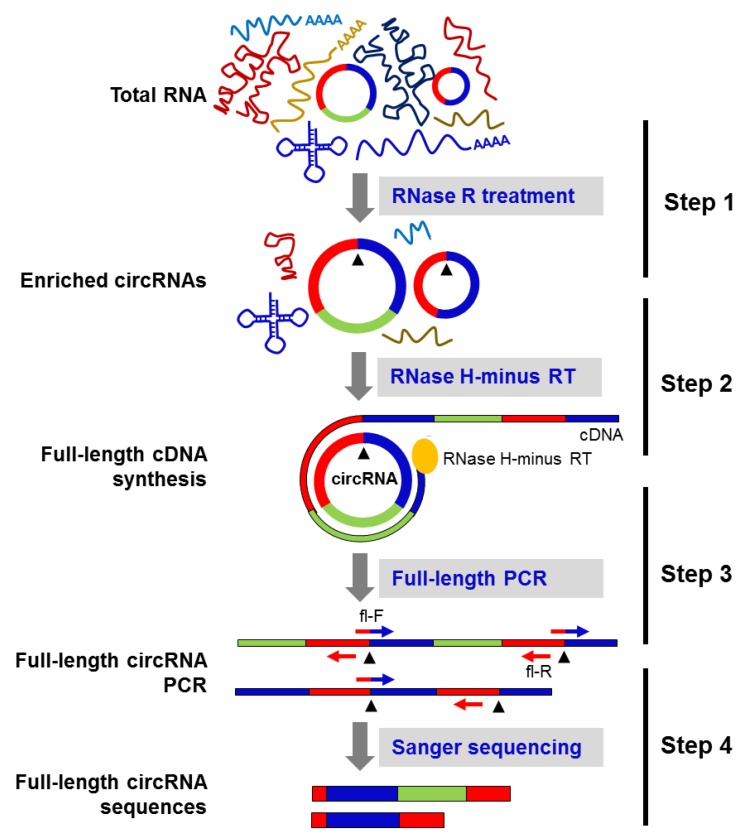
Schematic of the circRNA-Rolling Circle Amplification (circRNA-RCA) method. Step 1: total RNA is digested with RNase R to enrich the circRNA population. Step 2: the enriched circRNA pool is reverse-transcribed with RNase H-minus reverse transcriptase. Step 3: cDNA prepared in the above step is amplified by PCR, using full-length primers. Step 4: Sanger sequencing of the full-length circRNA PCR products identifies the mature spliced sequence of circRNAs and their splice variants. Black arrowheads represent the backsplice site.

**Figure 3 ijms-20-03988-f003:**
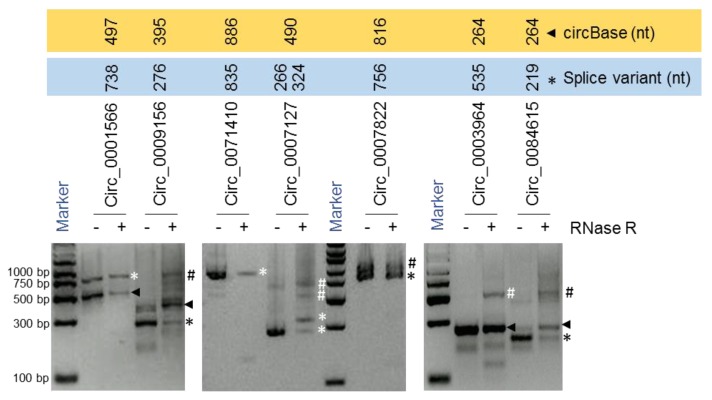
circRNA-RCA for full-length circRNA amplification. RT-PCR products amplified with full-length PCR primers and resolved on SYBR Gold-stained, 2% agarose gels. Arrowheads represent the sequence reported by circBase, while asterisks represent the splice variants. Hashtags represent either the doublets of circRNA or other circRNA isoforms or nonspecific products.

**Figure 4 ijms-20-03988-f004:**
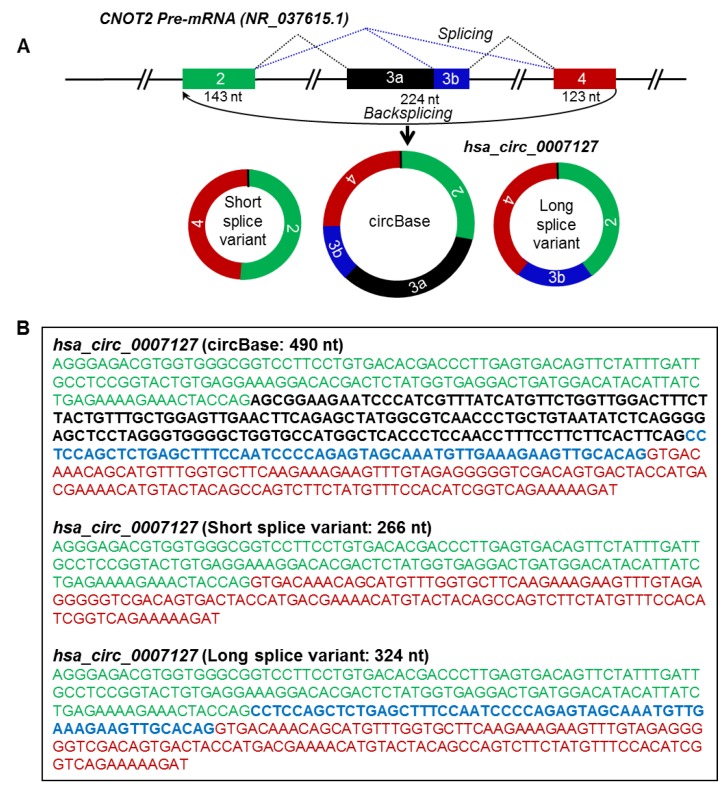
Alternative splicing generates circRNA splice variants. (**A**) Schematic representation of the *CNOT2* pre-mRNA and biogenesis of *hsa_circ_0007127* splice variants by alternative splicing. Boxes and straight lines represent exons and introns, respectively. The black dotted lines represent splicing, while the blue dotted lines represent alternative splicing. (**B**) Spliced full-length sequences of circBase and splice variants of *hsa_circ_0007127.* The text color matches the color of the exon box in panel A.

**Figure 5 ijms-20-03988-f005:**
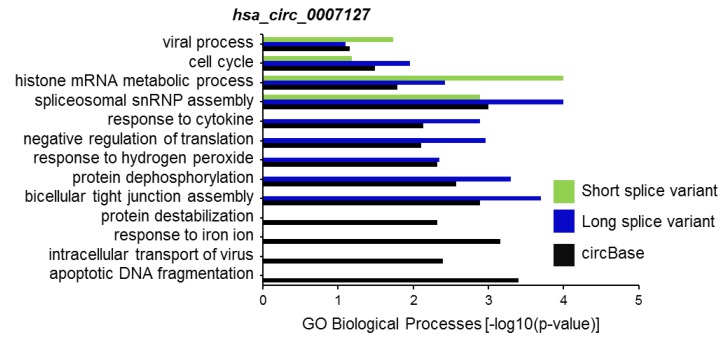
Selected list of biological processes enriched in Gene Ontology (GO) annotation analysis for the target genes of miRNAs associated with the sequence of circBase and splice variants of *hsa_circ_0007127*.

**Table 1 ijms-20-03988-t001:** circRNAs analyzed for splice variant expression in HeLa cells. The exon counts for each circRNA splice variant are indicated in parentheses.

CircRNA_junction_IDs (hg19)	circBase ID	Parent Gene	CircBase Length (Exon Count)	CircBase Length of Exons	Splice Variant Length (Exon Count)	Splice Variant Length of Exons
hsa_chr8_62593526_62596747_R	hsa_circ_0084615	*ASPH*	264 (3)	150, 45, 69	219 (2)	150, 69
hsa_chr12_70671911_70704797_F	hsa_circ_0007127	*CNOT2*	490 (3)	143, 224, 123	266 (2)	143, 123
					324 (3)	143, 58, 123
hsa_chr1_23356961_23385660_F	hsa_circ_0007822	*KDM1A*	816 (8)	166, 60, 134, 79, 93, 107, 82, 95	756 (7)	166, 134, 79, 93, 107, 82, 95
hsa_chr4_169812072_169837178_F	hsa_circ_0071410	*PALLD*	886 (7)	136, 99, 51, 222, 150, 95, 133	835 (6)	136, 99, 222, 150, 95, 133
hsa_chr15_90414706_90432372_R	hsa_circ_0009156	*AP3S2*	395 (4)	92, 112, 119, 72	276 (3)	92, 112, 72
hsa_chr5_179688683_179707608_R	hsa_circ_0001566	*MAPK9*	497 (4)	169, 130, 59, 139	738 (5)	169, 241, 130, 59, 139

**Table 2 ijms-20-03988-t002:** List of miRNAs predicted to associate with circBase and splice variant sequences of *hsa_circ_0007127* using the miRDB web tool. The number of binding sites for each miRNA is indicated in parentheses.

miRNA Targets of *hsa_circ_0007127* (Number of Binding Sites)		
Short Splice Variant (266 nt)	Long Splice Variant (324 nt)	circBase (490 nt)
hsa-miR-20b-3p (1)	hsa-miR-20b-3p (1)	hsa-miR-20b-3p (1)
hsa-miR-4261 (1)	hsa-miR-4261 (1)	hsa-miR-4261 (1)
hsa-miR-4463 (1)	hsa-miR-4463 (1)	hsa-miR-4463 (1)
hsa-miR-513a-5p (1)	hsa-miR-513a-5p (1)	hsa-miR-513a-5p (1)
hsa-miR-668-3p (1)	hsa-miR-668-3p (1)	hsa-miR-668-3p (1)
hsa-miR-6833-5p (1)	hsa-miR-6833-5p (1)	hsa-miR-6833-5p (1)
hsa-miR-4330 (1)	hsa-miR-4330 (1)	hsa-miR-4330 (1)
	hsa-miR-4487 (1)	hsa-miR-4487 (1)
	hsa-miR-4443 (1)	hsa-miR-4443 (1)
		hsa-miR-3929 (1)
		hsa-miR-4419b (1)
		hsa-miR-4478 (1)
		hsa-miR-766-3p (1)

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
