# Peer review of "Rolling Circle cDNA Synthesis Uncovers Circular RNA Splice Variants"

_ijms, 2019, doi:10.3390/ijms20163988_

Round 1
Reviewer 1 Report
Accumulating evidence indicates that circular RNAs are important molecules that play a role in diverse cellular processes. In this study, the authors reported a new method called circRNA-Rolling Circle Amplification (circRNA-RCA) that would be able to identify different splicing variants. This is a technically innovative study that has potential applications in circular RNA research field. Overall approach is straightforward. I just have a couple of minor concerns.
These potential splice variants could result from transcriptional noise; their biological relevance needs to be carefully characterized.
In Fig. 3, why there is no 490 bp band for Circ_007127?
Reviewer 2 Report
Manuscript Number: IJMS-558120
Title: Rolling Circle cDNA Synthesis Uncovers Circular RNA Splice Variants
In this manuscript, the authors report the development of a new method, termed circRNA-Rolling Circle Amplification (circRNA-RCA), to detect full-length circRNA sequences. The lead author has published quite extensively on original research and methodologies on circRNA analysis.
Using circRNA-RCA, different sizes of circRNA isoforms were obtained. One concern the authors need to address is that in Figure 3, multiple unspecified, and therefore presumptive nonspecific, bands seemed to have been generated by this method. Were these nonspecific bands also subjected to sequencing analysis? The authors needs to discuss what these nonspecific bands are, and whether nonspecific bands affects the overall analysis.
The MS may be improved on minor editing of the English expression.
Other minor comments:
The abstract is too long (272 words), exceeding the word limit of <200 words of the IJMS format.
The abbreviations should be explained more fully in the text.
The references do not seem to fit the IJMS format.
Some figures should be presented as tables. e.g. Figure 5, Fig S8 and Fig. S9
“sanger” should be capitalized to Sanger e.g. page 3, line 92; page 5, line 266.
In conclusion, the manuscript could be accepted for publication after addressing the minor comments by authors.
